# Mango-Stone-Derived Nitrogen-Doped Porous Carbon for Supercapacitors

**DOI:** 10.3390/mi13091518

**Published:** 2022-09-14

**Authors:** Yi Wang, Xinzi Yuan, Xingyu Guan, Kunling Ren, Yan Yang, Jun Luo, Yantao Zheng

**Affiliations:** 1College of Chemistry and Material Engineering, Guiyang University, Guiyang 550005, China; 2Mechanical College, Saint Petersburg State Technical University, 190013 Saint Petersburg, Russia; 3Xifeng Phosphorite Mine Co., Ltd., Guiyang 551100, China

**Keywords:** symmetric supercapacitor, mango stone, nitrogen-doped porous carbon

## Abstract

The preparation of N-doped porous carbon (NC-800) is presented via facile mango stone carbonization at 800 °C. The NC-800 material exhibits good cycle stability (the capacity retention is 97.8% after 5000 cycles) and high specific capacitance of 280 F/g at 1 A/g. Furthermore, the assembled symmetric device of NC-800//NCs-800 exhibits about 31.1 Wh/kg of energy density at 800 W/kg in a voltage range of 0–1.6 V. The results of the study suggest that NC-800 may be a promising energy storage material for practical application.

## 1. Introduction

With the depletion of natural resources and concerns for the environment, energy storage devices must be developed. Owing to high energy density, charging and discharging rate, and long cycling life, supercapacitors are frequently utilized as efficient electrical-energy-storage systems. In particular, supercapacitors have a power density that is 100 times greater than conventional batteries [1,2,3,4,5,6]. Owing to their large surface area, superior conductivity, and excellent thermal and chemical stabilities, carbon materials with varied microtextures and abundance are very appealing materials for the electrode of supercapacitors [7,8,9,10]. High-surface-area, low-cost porous carbon is extensively investigated as an electrode material for supercapacitors. Carbon-based materials have a specific capacitance proportional to their surface area because the charge is stored at the electrical double-layer surface of the electrode. However, the capacitance of porous carbon is generally low even at a high surface area (less than 250 F/g) [11,12,13]. Graphene, carbons derived from metal–organic frameworks, and templated and activated porous carbons are just a few examples of the many carbon nanomaterials that have been reported as efficient electrodes for supercapacitors. However, the high cost and limited availability of the resources of raw materials hinder the large-scale production of these carbons and their use in supercapacitors. For the future commercialization of supercapacitors, the search for economical and renewable raw materials and the use of simple techniques to produce effective carbon materials remain crucial [14,15].

The introduction of heteroatoms is an effective strategy for enhancing electrical conductivity and wettability. The inherent hydrophobicity of the surface of some carbon materials will hinder the penetration of electrolytes. After doping with heteroatoms, the physical and chemical characteristics of the surface can be improved, which can provide more active sites for the storage of electrolyte ions. Among different kinds of heteroatom doping, nitrogen doping has been widely used. Nitrogen-containing functional groups with electron donor characteristics can not only provide partial pseudocapacitance by participating in the Faraday reaction, but also improve the conductivity and wettability of materials, thus enhancing the electrochemical performance of materials. The high-nitrogen-content carbon material can be obtained by treating biomass or chemical substances. In addition, some high-nitrogen-content reagents such as urea are used as nitrogen sources for preparing nitrogen-doped carbon materials. Therefore, proper introduction of heteroatoms will improve the electrochemistry of materials [16,17,18,19,20].

In this work, mango stone is a readily accessible natural biomass resource capable of easy conversion into carbon materials at a low cost. In addition, the carbonization reaction is an effective way to convert various precursors into porous carbon materials. Hence, this work employed simple one-step carbonization with urea to convert mango stone powder to produce NC-800 for the supercapacitor electrode. NC-800 can reach 280 F/g and showed remarkable cycle stability with a capacity retention of 96% after 5000 cycles. The assembled supercapacitor showed energy density (31.1 Wh/kg) with power density of 800 W/kg, based on its electrochemical features; a worthwhile practical energy store facility is demonstrated here.

## 2. Materials and Methods

### 2.1. Characterization Methods

The walnut powder was purchased from farm product processing Co., Ltd. Acetylene black, acetylene black, polytetrafluoroethylene (PVDF), and N-methyl pyrrolidone were purchased from Sigma-Aldrich. KOH. All electrochemical tests were carried out on a CHI660 workstation in two-electrode configurations (Chenhua, Shanghai, China, 2017). The electrochemical performance of the carbon material was measured by CV at scan rates (5–100 mv/s) and GCD (galvanostatic charge–discharge) profiles were conducted with current densities between 0.5 and 5 A/g. Electrochemical impedance spectra were calculated in the 100 kHz to 0.01 Hz frequency range. The microstructures of the NC-800 were analyzed by field emission electron microscopy (FESEM, Hitachi, S4800, Tokyo, Japan, 2015). A Jobin-Yvon HR800 spectrophotometer was used to obtain the Raman spectra. The compositions of samples were analyzed by XPS (K-Alpha, Thermo Scientific, New York, NY, USA).

### 2.2. The Preparation Processes of C-800 and NC-800

1 g of mango stone powder and 4 g of KOH was heated in a tube furnace at 800 °C for 2 h in N_2_ at a rate of 5 °C/min. The carbon that was obtained was washed with HCl at a concentration of 2 mol/L, and the pH was adjusted to 7. The C-800 was obtained after 4 h of drying at 60 °C.

Firstly, 1.2 g urea was dissolved into 20 mL of distilled water. Subsequently, 100 mg of as-prepared C-800 was added into the solution, followed by vigorous stirring for 30 min. The mixture was dried by a lyophilizer to obtain NC-800 powder.

### 2.3. Electrochemical Measurements

The fabrication of the working electrodes was carried out via mixing of the prepared NC-800 or C-800 (80 wt%) with acetylene black (10 wt%) and polyvinylidene fluoride (PVDF, 10 wt%) binder. The mixed sample was fully stirred and ground in an agate mortar with ethanol to form a homogeneous paste followed by the coating of the resulting mixture onto the nickel foam substrate (1 cm × 1 cm). Approximately 4 mg of composites were coated onto the electrode. Using a standard three-electrode setup, the electrochemical characteristics of a single electrode were studied. The working electrodes consisted of Ni foams that had been coated multiple times with NC-800 composites; the counter electrode was made from platinum foil (1 cm × 1 cm) and the reference electrode was made from a saturated Hg/HgO electrode. Experiments were performed in an aqueous electrolyte (6 M KOH) at room temperature. The equations are as follows [4,5,6].
Cs = I · Δt/(m • ΔV)(1)
E = C • ΔV^2^/7.2 (2)
P = E/Δt (3)

I represents the discharge current (mA), t represents the discharge time (s), m represents the mass loading of the material (mg) and ΔV is the voltage window during the discharge process (V).

## 3. Results

### 3.1. Characterization

XPS survey spectra of NC-800 and C-800 were used to confirm the existence of C, O, and N. In Figure 1a, NC-800 exhibits predominant C 1s, O 1s, and N 1s peaks at 284, 533, and 400 eV, respectively (Figure 1a). The C1 spectra exhibit three signals at 284, 294, 296.2 eV. The N1s peak of NC-800 is located at 400.0 eV, corresponding to pyrrolic nitrogen in Figure 1b,c. Pyrrolic nitrogen provides pseudocapacitance, while nitrogen improves the conductivity of NC-800; The content of C, O, N is shown in Table 1.the Raman spectra of the NC-800 show two peaks appearing at 1340 and 1590 cm^−1^ belonging to the D- and G-band, respectively (Figure 1d). The D-band represented the disordered and defects of graphite while the G-band represents the configuration of crystalline graphite. The intensity ratio of D to G bands (ID/IG) shows the graphitization degree quantitatively. The peak ratio of NC-800 and C-800 is 0.84 and 0.87, respectively. The results show the graphitization degree of NC-800 is higher than that of C-800. From the SEM images with different magnification, the NC-800 also has a lot of nanopores on the surface (Figure 1e–g). As shown in Figure 1h, the BET results show that the surface area of NC-800 (85 cm^2^/g) is larger than that of C-800 (51 cm^2^/g).

### 3.2. Electrochemical Performances

The electrochemical performances of NCs-800 and C-800 for supercapacitors were estimated in KOH solution (6 M) using a standard three-electrode system. Typical CV curves of NC-800 at various scan rates over a potential range from −1 to 0 V (vs. Hg/HgO) are given in Figure 2a. However, the CV curves of NC-800 electrodes show a wide inverted hump and rectangular shape, showing pseudocapacitance and EDLCs capacitance due to N-doping. The GCD curves of NC-800 are as shown in Figure 2b; the capacitance (Cg) values of NC-800 are 280, 224, 135, 72, and 50 F/g at 1, 2, 5, 8, and 10 A/g, respectively. Together with the GCD results (Figure 2c), it can be concluded that NC-800(280 F/g) has a higher specific capacitance than that of C-800 (120 F/g) (Figure 2d). EIS was assessed at 0.01 Hz–100 kHz (Figure 2e). In the high-frequency region, the intercept at the real impedance (Z’) axis is related to the internal resistance. The vertical curve in the low-frequency area proves the fast ion diffusion behavior. The steeper slope means a smaller value of Warburg impedance. The value of the impedance (NC-800) is smaller than that of the C-800, which is ascribed to the introduction of N that improves the wettability. The NC-800 exhibits a good cycle life with 97.8% capacity reservation after 5000 cycles (Figure 2f).

An NC-800//NC-800 symmetric supercapacitor was fabricated for the practical application of the NC-800 electrode in supercapacitor devices. In Figure 3a, the shapes of the CV curves maintain their shape at various scan rates (2 to 100 mV/s) within the voltage window of 0–1.6 V. Figure 3b depicts the capacitance determined from GCD curves with current densities ranging from 1 to 10 A/g; the GCD curves exhibit a specific capacitance of 140, 116, 114, 64, and 30 F/g, and they retain the same shape, exhibiting good coulomb efficiency of the device. The symmetric supercapacitor exhibits about 31.1 Wh/kg of energy density at 800 W/kg of power density. In the Ragone plot of NC-800//NC-800 from Figure 3c, the diffusion resistance and internal resistance of the device are smaller, showing good conductivity. Moreover, as shown in Figure 3d, this device exhibits satisfactory cycle stability (89.3% capacity reservation after 5000 cycles). Based on this performance, our device outshines many reported carbon-based supercapacitors (Figure 3e and Table 2) [21,22,23,24,25,26,27,28]. The good electrochemical performance shows that the device based on NC-800//NC-800 is a promising practical energy-storage facility.

## 4. Conclusions

In conclusion, N-doped porous carbons were prepared using a simple procedure. The NC-800 exhibited around 280 F/g of specific capacitance at 1 A/g with 97.8% capacity retention and superb cycle stability after 5000 cycles. The symmetric supercapacitor of NC-800//NC-800 exhibited a special energy density of about 31.1 Wh/kg at 800 W/kg. The findings could lead the way for applications in the development of high-performance electrochemical devices made from inexpensive natural biomass.

## Figures and Tables

**Figure 1 micromachines-13-01518-f001:**
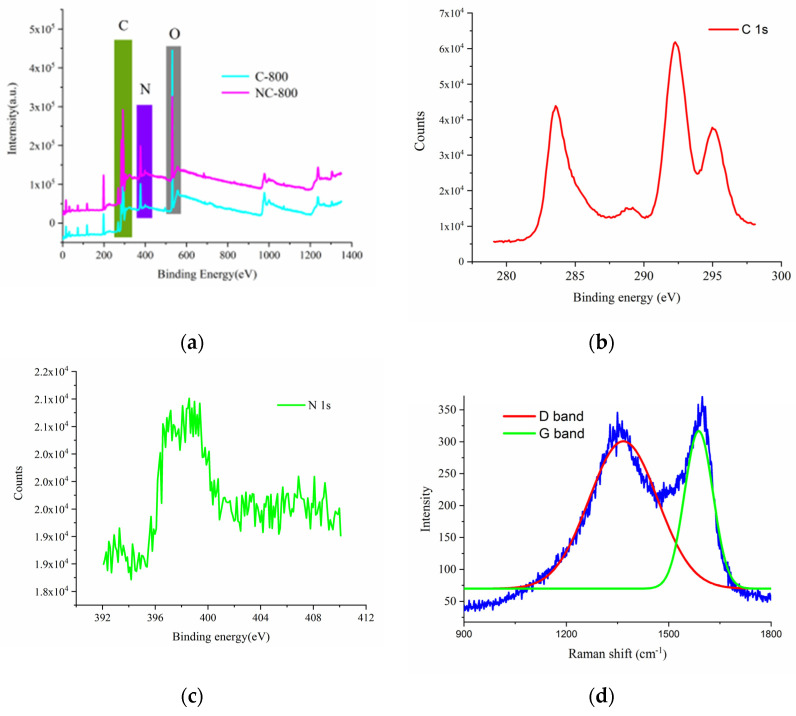
(**a**) XPS spectra; (**b**) Raman spectra of NC-800 and C-800; (**c**) C1s XPS spectra (**d**) N1s XPS spectra (**e**) SEM images of NC-800(200 nm); (**f**) SEM images of NC-800 (1 µm) (**g**) TEM images of NC-800 (**h**) BET of NC-800 and C-800.

**Figure 2 micromachines-13-01518-f002:**
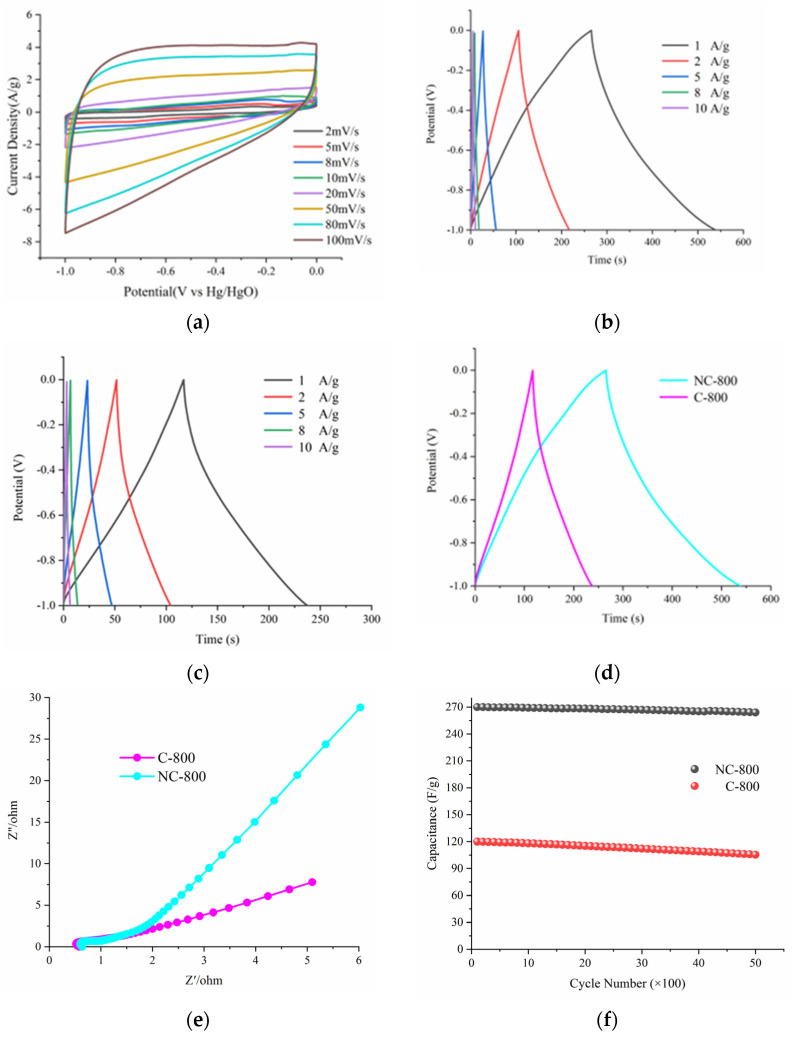
(**a**) CV curves of the NC-800 at different scan rates (2–100 mv/s); (**b**) GCD curves of NC-800 at different current densities (1–10 A/g); (**c**) GCD curves of C-800 at different current densities (1–10 A/g); (**d**) Comparison of NC-800 and C-800; (**e**) Nyquist plots of NC-800 and C-800; (**f**) Cycling performance of NC-800 and C-800 electrode for 5000 cycles.

**Figure 3 micromachines-13-01518-f003:**
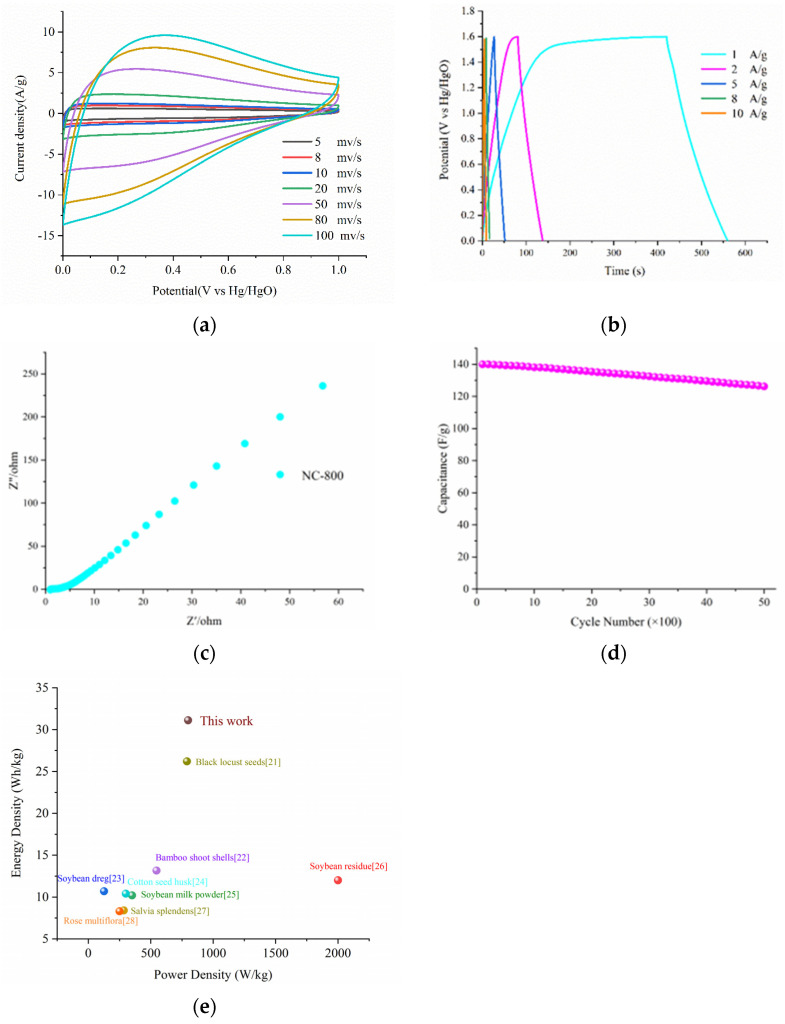
(**a**) CV curves of the NC-800//NC-800 at different scan rates (2-100 mv/s); (**b**) GCD curves of NC-800//NC-800 at different current densities (1-10 A/g); (**c**) Nyquist plots of NC-800//NC-800; (**d**) Cycling performance of NC-800//NC-800 electrode for 5000 cycles (**e**) Supercapacitor performance of plant-based heteroatom-doped biomass carbon electrodes.

**Table 1 micromachines-13-01518-t001:** Carbon, Nitrogen and Oxygen content of NC-800 and C-800.

	C Atomic%	N Atomic%	O Atomic%
NC-800	92.5	3.1	4.4
C-800	95.6	1.7	2.7

**Table 2 micromachines-13-01518-t002:** Supercapacitor performance of plant-based heteroatom-doped biomass carbon electrodes.

Biomass	E (Wh/kg)	P (W/kg)	Electrolyte
Black locust seeds	26.2	790	6 M KOH
Bamboo shoot shells	13.15	546.6	1 M NaSO_4_
Soybean dreg	10.7	125.6	6 M KOH
Cotton seed husk	10.4	300	6 M KOH
Soybean milk powder	10.2	351	6 M KOH
Salvia splendens	8.4	284.7	6 M KOH
Rose multiflora	8.3	250	6 M KOH

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
