# Peer review of "Mango-Stone-Derived Nitrogen-Doped Porous Carbon for Supercapacitors"

_micromachines, 2022, doi:10.3390/mi13091518_

Round 1

Reviewer 1 Report

In the work, authors report the mango stone derived N-doped porous carbon for supercapacitors. The unique carbon exhibits attracting electrochemical performance. The work is well prepared, however, necessary modifications are still needed as follows.

1.      The intrinsic nature of carbon materials plays a great role in enhanced capacitances. Thus corresponding fitting of Raman data (Fig. 1b) should be conducted. Please reference other reports (such as ACS Appl. Energy Mater. 2019, 2, 548; J. Mater. Chem. A 2022, 10, 2932) before.

2.      The upper voltage limit of the symmetric cells is about 1.6 V (Fig. 3a). Is it suitable for a symmetric device to stably operate? Besides, the low Columbic efficiency is obvious in Fig. 3b, why?

3.      How to obtain the specific capacitance, energy density and power density? Please specify the equations applied in the work. Additionally, the mass loading of active materials in three-electrode and/or two-electrode systems should be provided for better understanding the electrochemical properties here.

4.      Corresponding comparisons in electrochemical behaviors of the carbon electrodes should be carried out with other reported biomass-derived carbon electrodes.

Author Response

Dear editor,

Thank you very much for your letter and the referees’ reports. Based on your comment and request, we have made extensive modification on the original manuscript. Here, we attached revised manuscript. A document answering every question from the referees was also summarized and enclosed. A revised manuscript with the correction sections red marked was attached as the supplemental material and for easy check/editing purpose.

Should you have any questions, please contact us without hesitate.

Reviewer 1

In the work, authors report the mango stone derived N-doped porous carbon for supercapacitors. The unique carbon exhibits attracting electrochemical performance. The work is well prepared, however, necessary modifications are still needed as follows.

  1. The intrinsic nature of carbon materials plays a great role in enhanced capacitances. Thus corresponding fitting of Raman data (Fig. 1b) should be conducted. Please reference other reports (such as ACS Appl. Energy Mater. 2019, 2, 548; J. Mater. Chem. A 2022, 10, 2932) before.

Thanks for your Suggestion. The corresponding fitting of Raman data is added in the manuscript.

  1. The upper voltage limit of the symmetric cells is about 1.6 V (Fig. 3a). Is it suitable for a symmetric device to stably operate? Besides, the low Columbic efficiency is obvious in Fig. 3b, why?

Thanks for your question. The stable voltage region is in the range of 0–1.8 V in the related reference (Applied Surface Science 2021, 566, 150613). The repeated expansion and contraction of the electrode material during the repeated charge and discharge process cause the electrode material to be powdered and separated from the current collector, resulting in a reduction in capacity and coulomb efficiency.

  1. How to obtain the specific capacitance, energy density and power density? Please specify the equations applied in the work. Additionally, the mass loading of active materials in three-electrode and/or two-electrode systems should be provided for better understanding the electrochemical properties here.

Thanks for your suggestion. The equations is as followed.

Cs = I · Δt/(m • ΔV)     (1)

E = C • ΔV2 /7.2        (2)

P = E/Δt              (3)

I represents the discharge current (mA), t represents the discharge time (s), m represents the mass loading of the material (mg) and ΔV is the voltage window during discharge process (V), respectively (Applied Surface Science, 2020,510,145384).

  1. Corresponding comparisons in electrochemical behaviors of the carbon electrodes should be carried out with other reported biomass-derived carbon electrodes.

Thanks for your suggestion. The electrochemical behaviors of the carbon electrodes are added in the manuscript.

Biomass

E (Wh/kg)

P(W/kg)

Electrolyte

Black locust seeds

26.2

790

6 M KOH

Bamboo shoot shells 

13.15

546.6

1 M NaSO4

Soybean dreg

10.7

125.6

6 M KOH

Cotton seed husk

10.4

300

6 M KOH

Soybean milk powder

10.2

351

6 M KOH

Salvia splendens

8.4

284.7

6 M KOH

Rose multiflora

8.3

250

6 M KOH

Reviewer 2 Report

N-doped porous carbon is prepared by KOH-activation of Mango Stone for supercapacitor electrode material. After treating by urea, the supercapacitive performance is significantly enhanced. This work should be improved before acceptance.

1. Materials analysis: N2 sorption, TEM should be carried out. 

2. XPS spectra, C1s, N1s, O1s, and the species contents should be given.

3. The influence of urea treatment on the surface chemistry and supercapacitive performance should be deeply discussed.

4. The voltage window of the two electrode system could not exceed 1.0V. Fig 3a shows obvious the redox peak of H2O decomposition.

Author Response

Dear editor,

Thank you very much for your letter and the referees’ reports. Based on your comment and request, we have made extensive modification on the original manuscript. Here, we attached revised manuscript. A document answering every question from the referees was also summarized and enclosed. A revised manuscript with the correction sections red marked was attached as the supplemental material and for easy check/editing purpose. Should you have any questions, please contact us without hesitate.

Reviewer 2

N-doped porous carbon is prepared by KOH-activation of Mango Stone for supercapacitor electrode material. After treating by urea, the supercapacitive performance is significantly enhanced. This work should be improved before acceptance.

  1. Materials analysis: N2 sorption, TEM should be carried out.

Thanks for your suggestion. The BET and TEM is added.

  1. XPS spectra, C1s, N1s, O1s, and the species contents should be given.

Thanks for your suggestion.

C  Atomic %

N  Atomic %

  O  Atomic %

 NC-800

92.5

       3.1

       4.4

  C-800

        95.6

       1.7

       2.7

  1. The influence of urea treatment on the surface chemistry and supercapacitive performance should be deeply discussed.

Thanks for your suggestion. The influence of urea treatment on the surface chemistry and supercapacitive performance is added.

The inherent hydrophobicity of the surface of some carbon materials will hinder the penetration of electrolyte. After doping with heteroatoms, the physical and chemical characteristics of the surface can be improved, which can provide more active sites for the storage of electrolyte ions. Among different kinds of heteroatom doping, nitrogen doping has been widely used. Nitrogen-containing functional groups with electron donor characteristics can not only provide partial pseudocapacitance by participating in Faraday reaction, but also improve the conductivity and wettability of materials, thus enhancing the electrochemical performance of materials. The high nitrogen content carbon material can be obtained by treating biomass or chemical substances. In addition, some high content nitrogen reagents such as urea are used as nitrogen sources for preparing nitrogen doped carbon materials. Therefore, proper introduction of heteroatoms will improve the electrochemistry of materials.

  1. The voltage window of the two electrode system could not exceed 1.0V. Fig 3a shows obvious the redox peak of H2O decomposition.

Thanks for your suggestion. The voltage window is revised as 0-1.0V.

Round 2

Reviewer 2 Report

1. C1s and N1s should be added in the maintext and described.

Author Response

Thanks for your suggestion. The C1s and N1s is added in the maintext.